# Two Heads are Better than One: Retrieval Augmented LLM for Question Answering with External Knowledge Attention

## Abstract

Retrieval-augmented generation (RAG) of large language models (LLMs) has recently attracted significant attention owing to their ability to address knowledge gaps in generating reliable answers for specific questions. Existing RAG approaches typically optimize the knowledge processing by filtering out irrelevant or incorrect information and restructuring it for model input, improving the accuracy of answers to given questions. A general approach in doing so is to combine the retrieved knowledge with the input inquiry, which are then fed into the LLM to produce an answer. This approach requires the LLM to have strong knowledge comprehension and reasoning capabilities to effectively utilize the useful information, which may lead to errors when it fails to correctly interpret relevant knowledge. In this paper, we propose a novel approach to augmenting LLMs with external knowledge attention for question answering (QA), where the attention is functionalized as an extra head that integrated with the internal heads used in LLMs. We develop a memory-based mechanism that dynamically controls the degree of knowledge integration with the extra head based on the relationship between the question and the retrieved knowledge, and allows for differentiated fusion of external knowledge and LLM ability at its different layers. Experiments on both general and specific-domain QA tasks demonstrate the effectiveness of our approach, highlighting its potential to optimize LLMs for similar challenges.

## 1 Introduction

Recently, large language models (LLMs) (Ouyang et al., 2022; Touvron et al., 2023; Chiang et al., 2023; OpenAI, 2023; Taori et al., 2023) have achieved remarkable success in artificial intelligence (AI). One essential reason for this effectiveness is the capability of LLMs to learn various types of knowledge—such as linguistic structures and commonsense reasoning—from large-scale labeled and unlabeled data. Therefore, LLMs follow human instructions to complete a wide range of tasks, particularly question-answering (QA). However, when relying solely on the knowledge learned by the LLM from massive data is insufficient to generate reliable answers, the performance of LLMs always degrade. Such situation constantly arises with the rapidly evolving or highly specialized knowledge, making it challenging for the LLM to provide accurate and up-to-date responses. To this end, integrating new knowledge becomes crucial to help LLMs generating appropriate answers for input questions.

To address the aforementioned situation, retrieval-augmented generation (RAG) (Baek et al., 2023b; Sun et al., 2023; Asai et al., 2023; Xiong et al., 2024) has been proven to be a feasible solution, which generally retrieve knowledge relevant to the input from existing knowledge bases or documents and use this information to instruct LLMs in generating responses. Because the results of knowledge retrieval may contain noise or irrelevant information, most studies focus on further optimizing the retrieved knowledge to improve RAG performance (Xiong et al., 2024; Fang et al., 2024; Shi et al., 2024). For example, Zhu et al. (2024) propose a mechanism to filter out content in the retrieval results that is irrelevant or misleading to current questions; Xu et al. (2024) design a rewriting module to refine the retrieved knowledge. These approaches generally apply a string concatenation operation to the retrieved knowledge and the LLM's input (e.g., the question), resulting in a less converged integration of knowledge and LLMs, so that requiring LLMs to have a strong understanding

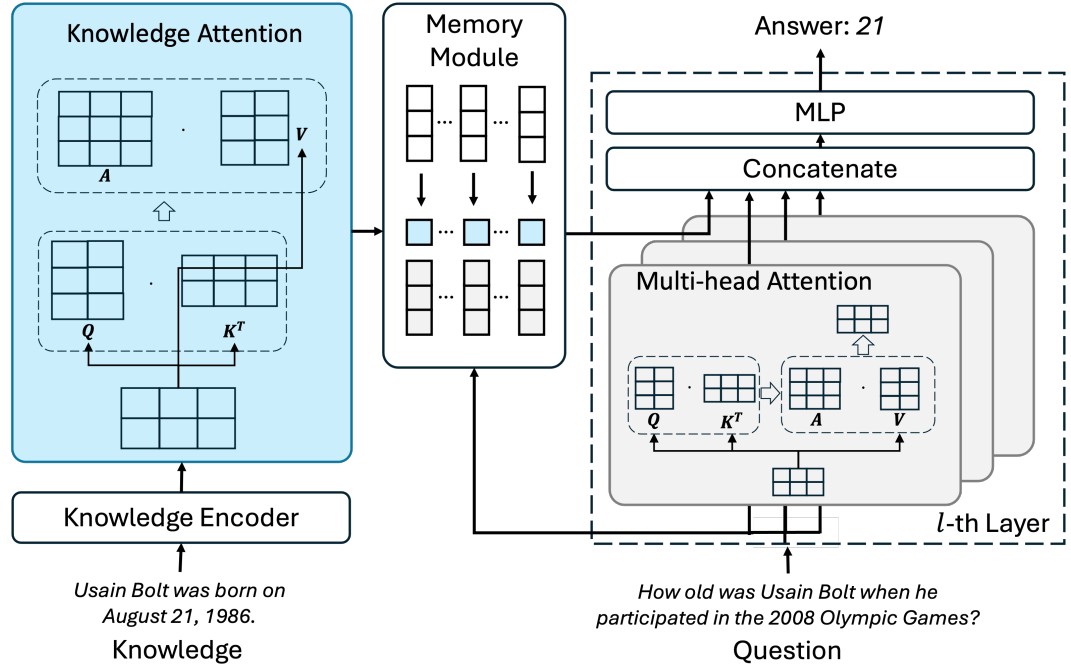

Figure 1: The overall architecture of our approach for QA. The left part illustrates the knowledge attention to encode knowledge; the middle presents the memory module to assign weights to the encoded knowledge for various LLM layers; the right shows the $l$-th layer in the LLM.

and commonsense reasoning abilities to effectively utilize the knowledge. Consequently, there is a risk that errors in the process of understanding or reasoning may lead to incorrect usage of knowledge. Consider that existing research in text representation learning indicates that using vectorized representations have better ability in fusing different pieces of information (Pennington et al., 2014; Devlin et al., 2019a; Brown et al., 2020; Ouyang et al., 2022), it is expected to have an advanced augmentation strategy for LLMs with an appropriate form of knowledge inputs.

In this paper, we propose a novel RAG-based approach for QA tasks, which vectorizes the retrieved knowledge and deeply integrates it into the decoding process of the LLM. Specifically, we propose an external knowledge attention mechanism within the multi-head attention of each LLM layer, which is designed to fuse the encoded knowledge with the model's representations at different layers, enabling more effective utilization of external information. Furthermore, we propose a memory-based mechanism that dynamically controls the degree of knowledge fusion based on the relationship between the question and the retrieved knowledge, which allows for differentiated and deeper knowledge integration at various layers, enhancing the LLM's capability to generate accurate answers. We run experiments on English benchmark QA datasets in the general and medical domain, where our approach outperforms existing studies and demonstrates its effectiveness in improving model performance by efficiently integrating essential knowledge.

## 2 THE APPROACH

Given an input question $\mathcal{X}$ and the extracted knowledge $\mathcal{S}$, our approach aims to generate the relevant answer $\mathcal{Y}$. To achieve this, we employ a knowledge attention mechanism $f_{KA}$ to encode the retrieved knowledge. The knowledge attention serves as an additional head within the multi-head attention framework of each layer in the LLM. This specialized head processes the encoded knowledge representations and integrates them with the outputs of the other attention heads, thereby facilitating a deep fusion of knowledge and the LLM. This mechanism enables the model to effectively combine external knowledge, enhancing its ability to generate accurate and contextually relevant responses. Additionally, we utilize a memory module $f_M$ to compute the weights for integrating knowledge across different layers of the LLM. The memory module determines the significance of

each knowledge instance at each layer, allowing for differentiated and nuanced integration of various information. The overall architecture of our approach is illustrated in Figure **??**, where the left side presents the knowledge attention and the right side highlights the memory module as well as the LLM for answer generation. Therefore, our approach is formally represented as:

$$\mathcal{Y} = f_{K-LLM}(\mathcal{X}, f_M(f_K(\mathcal{S}))) \tag{1}$$

In the following sections, we first introduce the knowledge attention mechanism, followed by the memory module, and finally describe the knowledge fusion approach with knowledge attention.

## 2.1 Knowledge Attention

The purpose of the knowledge attention mechanism is to fuse the encoded knowledge with the LLM, thereby guiding it to generate reliable and accurate answers. Consider that different knowledge instances may contribute variably to the generation of responses, we adopt a weighted approach to dynamically and selectively utilize different pieces of knowledge. This strategy minimizes the impact of potentially irrelevant or noisy information on the final answer. The knowledge attention mechanism comprises two main components: a knowledge encoder and an attention module. The knowledge encoder $f_{KA}$ is responsible for encoding the retrieved knowledge $\mathcal{S}$, which typically consists of a diverse set of knowledge instances that may vary in their representation formats, such as full-text articles or multiple-word entities. For knowledge instances presented in article or paragraph form, we directly utilize a pre-trained encoder (e.g., BERT (Devlin et al., 2019b)) to encode the text. This leverages the encoder's strong capabilities in capturing semantic and contextual information from continuous text. For knowledge instances consisting of multiple words or entities, we first concatenate these words into a coherent sentence. This concatenated sentence is then passed through the same pre-trained encoder to obtain a unified representation. Since these knowledge instances do not possess an inherent sequential order, we omit positional embeddings during the encoding process.

Specifically, for each knowledge instance $s_u \in \mathcal{S}$, the encoding process is as follows:

$$\mathbf{h}_u^S = f_{KA}(s_i) \tag{2}$$

where $\mathbf{h}_u^S \in \mathbb{R}^d$ represents the encoded vector of the $u$-th knowledge instance, and $d$ is the dimensionality of the encoder's hidden states. By encoding each knowledge instance individually, we obtain a set of vector representations:

$$\mathcal{H}^S = \{\mathbf{h}_1^S, \mathbf{h}_2^S, \ldots, \mathbf{h}_U^S\} \tag{3}$$

where $U$ is the number of knowledge instances.

Once the knowledge instances are encoded, the attention module processes these representations to determine their relevance and importance in the context of the input question $\mathcal{X}$. The attention module operates as follows: We project the encoded knowledge matrix $\mathbf{H}^S$ into query $\mathbf{Q}^S$, key $\mathbf{K}^S$, and value $\mathbf{V}^S$ matrices using learnable projection matrices $\mathbf{W}^{S,Q}$, $\mathbf{W}^{S,K}$, and $\mathbf{W}^{S,V}$, respectively:

$$\mathbf{Q}^S = \mathbf{H}^S\mathbf{W}^{S,Q}, \quad \mathbf{K}^S = \mathbf{H}^S\mathbf{W}^{S,K}, \quad \mathbf{V}^S = \mathbf{H}^S\mathbf{W}^{S,V} \tag{4}$$

We compute the attention weights $\mathbf{A}^S$ using the scaled dot-product attention mechanism:

$$\mathbf{A}^S = \text{softmax}\left(\frac{\mathbf{Q}^S(\mathbf{K}^S)^\top}{\sqrt{d_k}}\right) \tag{5}$$

where $d_k$ is the dimensionality of the key vectors. The attention weights are then applied to the value matrix to obtain the weighted knowledge representations $\mathbf{H}'^S$:

$$\mathbf{H}'^S = \mathbf{A}^S\mathbf{V}^S \tag{6}$$

## 2.2 Memory-based Knowledge Head Weighting

Building on the observation that different layers of LLMs process distinct types of information, we propose a memory module to determine the optimal layers for integrating external knowledge.

Intuitively, for various questions, the parameters that activate the relevant knowledge to generate accurate answers should vary, influenced by the depth of the LLM layers. To achieve this, our memory module assigns different weights to the integration of knowledge at each layer, ensuring that the most pertinent knowledge is utilized effectively where it is most relevant within the model's architecture. The memory module employs multiple memory vectors, each corresponding to a specific layer of the LLM. Let $\mathbf{m}_l$ denote the memory vector for the $l$-th layer, where $l = 1, 2, \ldots, L$ and $L$ is the total number of layers in the LLM. These memory vectors encapsulate the characteristics of knowledge integration specific to each layer, enabling the module to assess the relevance of each knowledge instance dynamically. To calculate the integration weight for a knowledge instance $s_n$ at layer $l$, firstly, we compute the average vector of the knowledge instance's representation $\mathbf{h}_n^S$ with all input vectors at layer $l$. This average vector captures the contextual alignment between the knowledge instance and the information being processed at that layer. Second, we calculate the cosine similarity $s_{l,n}$ between this average vector and the corresponding memory vector $\mathbf{m}_l$:

$$s_{l,u} = \frac{\mathbf{h}_u^S \cdot \mathbf{m}_l}{\|\mathbf{h}_u^S\| \cdot \|\mathbf{m}_l\|} \tag{7}$$

This cosine similarity score $s_{l,n}$ quantifies the degree to which the knowledge instance $s_n$ is relevant to the information processed at layer $l$. A higher similarity indicates a stronger relevance, suggesting that the knowledge instance should be more heavily integrated at that layer. Subsequently, we multiply each knowledge instance's vector $\mathbf{h}_u^S$ by its corresponding similarity score $s_{l,u}$, effectively weighting the knowledge instances based on their relevance to the specific layer. These weighted vectors are then concatenated to form the knowledge matrix $\mathbf{H}_l^S$ for layer $l$:

$$\mathbf{H}_l^S = \begin{bmatrix} s_{l,1}\mathbf{h}_1^S, & s_{l,2}\mathbf{h}_2^S, & \ldots, & s_{l,U}\mathbf{h}_U^S \end{bmatrix} \tag{8}$$

The matrix $\mathbf{H}_l^S$ encapsulates the weighted knowledge information tailored for integration at layer $l$. This matrix is then processed through Eq. (9), allowing it to be seamlessly fused with the LLM's representations at the corresponding layer.

## 2.3 RETRIEVAL AUGMENTATION FOR LLM WITH N+1 HEADS

After encoding the retrieved knowledge, we integrate it into the LLM by treating the knowledge attention as an additional knowledge head, which operates alongside the existing heads in each layer's multi-head attention module of the LLM, separately processing the encoded knowledge and the input question before fusing their outputs. The motivation for introducing an extra head stems from the design of multi-head attention, where each head captures different aspects of the input (Vaswani et al., 2017). Knowledge can be viewed as an implicit facet of the input, allowing it to be encoded alongside other aspects of the input question. This alignment with the original intent of multi-head attention facilitates a natural fusion of knowledge with the LLM.

Specifically, in the $l$-th layer, we employ the standard multi-head attention procedure by concatenating the output from the knowledge head, $\mathbf{H}_l'^S$, with the outputs from the standard heads. The process is formulated as

$$\mathbf{H}_l^X = \text{Norm}\left(\text{FFN}\left(\mathbf{H}_l'^S \oplus \left(\mathbf{H}_{l-1,1}^X \oplus \mathbf{H}_{l-1,1}^X \oplus \cdots \oplus \mathbf{H}_{l-1,N}^X\right)\right)\right) \tag{9}$$

where $\mathbf{H}_{l-1,1}^X \cdots \mathbf{H}_{l-1,N}^X$ are the output of the $N$ heads in the LLM. This concatenated matrix is then passed through the transformer's feed-forward and normalization layers to produce the output $\mathbf{H}_l^X$, which serves as the input to the $(l+1)$-th layer. We perform the same process for all layers and obtain the $\mathbf{H}_L^X$ for the last layer. Finally, we feed the last vector in $\mathbf{H}_L^X$ into a softmax classifier to predict the tokens in the answer.

# 3 EXPERIMENT SETTINGS

## 3.1 DATASETS

Following previous studies, we evaluate our approach on knowledge-intensive reasoning tasks. We utilize five question-answering (QA) datasets: CWQ (Talmor & Berant, 2018), GrailQA (Gu et al., 2021), QALD10 (Perevalov et al., 2022), WebQSP (Yih et al., 2016), and Simple Questions (Bordes

Table 1: The statistics of the datasets used in the experiments. "SQ" and "CF" are abbreviations for Simple Questions and CounterFact datasets.

| | CWQ | GRAILQA | QALD10 | WEBQSP | SQ | MEDMCQA | MEDQA | PUBMEDQA | CF |
|---|---|---|---|---|---|---|---|---|---|
| TRAIN | 1.8K | 53.8K | 0.4K | 2.6K | 19.5K | 182.8K | 10.2K | 24.6K | 17.9K |
| DEV | 2K | 7.0K | - | - | 2.8K | 4.2K | 1.3K | 1.5K | 2.0K |
| TEST | 2K | 3.5K | 0.4K | 1.4K | 5.6K | 6.2K | 1.3K | 1.5K | 2.0K |

et al., 2015), where the last one is a single-hop QA dataset and the other four are multi-hop QA datasets. In addition, to evaluate our approach in broader contexts, we include QA datasets from the medical domain and one dataset with counterfact settings (i.e., the CounterFact dataset Meng et al. (2022)), where we utilize four medical datasets, namely, MedMCQA (Pal et al., 2022), MedQA (Jin et al., 2020), and PubMedQA (Jin et al., 2019). Following existing studies (Meng et al., 2022; Das et al., 2024), we use the first 2,000 instances in CounterFact as the test set, and the 90% and 10% of the remaining ones as the training and development sets, respectively. For other datasets, we use the official train/dev/test splits of them. For multilingual datasets, we only use the English part. The number of examples in train/dev/test is illustrated in Table 1.

For the knowledge base, for general domain QA datasets, we use the WikiData knowledge base (Vrandečić & Krötzsch, 2014); for the medical datasets, we follow Xiong et al. (2024) and utilize the combination of PubMed[1], StatPearls[2], medical Textbooks (Jin et al., 2020), and Wikipedia[3] as the knowledge base. To perform models on the CounterFact, we regard all CounterFactual statements in the dataset as the knowledge base and prompt LLM to perform text completion to finish the task.

## 3.2 BASELINES

In the experiments, we run three baselines, namely, "**Base**", "**+RAG**", "**+RAG+M**". "**Base**" denotes the approach where we directly fine-tune the LLM on the training data without using RAG. "**+RAG**" utilizes RAG when performing the task, where the knowledge instances are directly concatenated with the input text to instruct the LLM to produce the response. The LLM is also fine-tuned in this case. "**+RAG+M**" refers to the approach that utilizes the knowledge encoder and the memory module (M) in our approach to perform the task, where the knowledge representation is directly concatenated with the output of each multi-head attention layer and the resulting matrix is fed into the next layer to process following the standard process. Compared with our approach, this approach does not utilize the knowledge attention to encode the knowledge. Following the notation, our approach is denoted as "**+RAG+M+KA**".

## 3.3 IMPLEMENTATION DETAILS

Since a good text representation is highly important to achieve outstanding performance for many NLP tasks, in the experiments, we use well-known pre-trained LLMs for the knowledge encoder and the LLM. Specifically, we use the base version of BERT Devlin et al. (2019a) and the chat version of LLaMA-2 Touvron et al. (2023), as the knowledge encoder and the LLM, respectively. For both models, we follow their default hyper-parameter settings. The BERT model contains 12 layers of multi-head attention (MHA) with 768-dimensional hidden vectors; For LLaMA-2, we use the 7B and 13B versions, where the 7B version has 32 layers of MHA with 4,096-dimensional hidden vectors, and the 13B version contains 40 layers of MHA with 5,120-dimensional hidden vectors.

To obtain the knowledge, for general domain QA and CounterFact datasets, we follow Sun et al. (2023) to extract the knowledge instances for different input text. For the medical domain QA datasets, we follow Xiong et al. (2024) to employ MedCPT[4] (Jin et al., 2023) to extract knowledge instances since they achieve good performance in the experiments. In evaluation, we use F1 for GrailQA, QALD10, and WebQSP; we employ the accuracy as the metric for all other datasets; The size of the retrieved knowledge instance is set to 10. We try the combinations of different hyper-

---

[1] https://pubmed.ncbi.nlm.nih.gov/

[2] https://www.statpearls.com/

[3] https://huggingface.co/datasets/wikipedia

[4] https://huggingface.co/ncbi/MedCPT-Query-Encoder

Table 2: The average performance of different models on the general domain QA

| APPROACH | CWQ | GRAILQA | QALD10 | WEBQSP | SQ |
|---|---|---|---|---|---|
| LLaMA-2 (7B) | 64.0 | 75.8 | 48.4 | 77.0 | 75.7 |
| + RAG | 65.2 | 77.1 | 49.6 | 78.1 | 76.7 |
| + RAG + M | 66.4 | 78.2 | 50.4 | 79.1 | 77.6 |
| + RAG + M + KA | 67.6 | 79.2 | 51.7 | 80.0 | 78.4 |
| LLaMA-2 (13B) | 68.0 | 79.5 | 52.2 | 80.7 | 79.4 |
| + RAG | 69.0 | 81.1 | 53.4 | 82.0 | 80.5 |
| + RAG + M | 70.2 | 82.1 | 54.7 | 82.8 | 81.4 |
| + RAG + M + KA | **71.6** | **83.3** | **56.0** | **84.1** | **82.7** |

Table 3: Model performance on medical domain QA and CounterFact dataset.

| Approach | MedMCQA | MedQA | MMLU-Med | PubMedQA | CF |
|---|---|---|---|---|---|
| LLaMA-2 (7B) | 53.0 | 68.4 | 67.1 | 65.5 | 59.7 |
| + RAG | 54.2 | 69.8 | 68.3 | 66.4 | 60.8 |
| + RAG + M | 55.0 | 70.4 | 69.2 | 67.6 | 61.5 |
| + RAG + M + KA | 56.0 | 71.2 | 70.1 | 68.3 | 62.5 |
| LLaMA-2 (13B) | 55.2 | 69.6 | 68.3 | 66.7 | 61.8 |
| + RAG | 56.3 | 70.8 | 69.6 | 67.7 | 62.9 |
| + RAG + M | 57.2 | 71.6 | 70.2 | 68.4 | 63.7 |
| + RAG + M + KA | **57.8** | **72.3** | **71.1** | **69.1** | **64.2** |

parameters for each model and choose the one achieving the highest accuracy on the development set in the final experiments. We run each model three times with different random seeds and report the average and standard deviation. In training, all parameters are updated, including the ones in BERT and LLaMA.

## 4 RESULTS AND ANALYSIS

### 4.1 OVERALL RESULTS

We run experiments with baselines and our approach using LLaMA-2 (7B) and LLaMA-2 (13B) on the benchmark datasets. The average performance of three runs of different models are reported in Table 2 and Table 3, where the former table presents the results in the general domain and the latter shows the results in the medical and CounterFactual datasets. There are several observations.

First, compared with the vanilla model, the one with RAG (i.e., "+ RAG") achieves better performance, which indicates that the vanilla LLaMA models require extra knowledge to appropriately answer these questions. Second, when the memory module is added on top of RAG, consistent improvements are observed on all datasets. These results indicate that it is essential to distinguish the contribution of different knowledge instances to the task. Third, our approach with RAG, memory, and the gate achieves the best performance on all datasets, which indicates the effectiveness of our approach to working with various pre-trained LLMs. The observation further confirms that the memory module and the gate mechanism work with each other. And it is crucial to not only distinguish the noise in the knowledge instances but also control the knowledge that is integrated into the LLM.

We further compare our best-performing model with previous studies on the test set of different datasets. We report the results on general domain QA in Table 4 and present the results on the medical domain QA in Table 5. The approaches that use LLM and RAG are marked by † and ‡, respectively, and the approaches fine-tuned on the training set are marked by ∗. For the medical QA, we also present the performance of in-domain LLM (e.g., Med-PaML 2 (Singhal et al., 2023)) for reference. It is observed that our approach outperforms most previous studies with general domain LLMs on most datasets. Particularly, our approach outperforms previous studies (Sun et al., 2023;

Table 4: Comparison with existing studies on general domain QA. ∗ marks the approach where the model is optimized on the training data. † marks the approach uses LLM. ‡ marks the approach that uses non-open-sourced LLMs (e.g., GPT-4).

| Approach | CWQ | GrailQA | QALD10 | WebQSP | SQ |
|---|---|---|---|---|---|
| ∗Das et al. (2021) | 70.0 | - | - | 76.1 | - |
| ∗Borroto et al. (2022) | - | - | 45.4 | - | - |
| †Yu et al. (2022) | 70.4 | 78.7 | - | 78.8 | - |
| Shu et al. (2022) | - | 75.3 | - | 78.9 | - |
| ∗†Gu et al. (2023) | - | 81.7 | - | 79.6 | - |
| ∗Baek et al. (2023a) | - | - | - | 65.3 | 85.8 |
| †‡Sun et al. (2023) | 69.5 | 81.4 | 53.8 | 82.6 | 66.7 |
| Ours | 71.6 | 83.3 | 56.0 | 84.1 | 82.7 |

Table 5: Comparison with existing studies on medical domain QA. ∗ marks the approach where the model is optimized on the training data. † marks the approach uses RAG. ‡ marks the approach that uses non-open-sourced LLMs (e.g., GPT-4).

| Approach | MedMCQA | MedQA | MMLU-Med | PubMedQA |
|---|---|---|---|---|
| †Singhal et al. (2023) | 72.3 | 86.5 | - | 81.8 |
| Touvron et al. (2023) | 36.3 | 35.2 | 46.3 | - |
| †‡Jeong et al. (2024) | 44.0 | 48.6 | 57.2 | - |
| †‡Xiong et al. (2024) | 58.0 | 67.4 | 75.5 | 67.8 |
| Ours | 57.8 | 72.3 | 71.1 | 69.1 |

Jeong et al., 2024) that leverage RAG to enhance general domain LLMs with the knowledge to answer the questions. This further presents that using the memory plugin and the gate module to selectively leverage the knowledge is important for producing better results. Besides, our approach fails to outperform the medical domain LLMs (e.g., Med-PaML 2 (Singhal et al., 2023)) on the medical QA datasets since these models are trained and tuned on large-scale medical data and they are able to learn the medical knowledge directly and thus show better performance.

## 4.2 THE EFFECT OF KNOWLEDGE ATTENTION

To explore the role of the Knowledge Attention mechanism, we conducted experiments using LLaMA-2 (13B) under the following settings: (1) **Equal Attention Weights**: All attention weights are assigned the same value, effectively removing the attention mechanism. In this setup, all knowledge instances are treated equally without distinction. (2) **Without Knowledge Attention**: Instead of constructing an additional knowledge attention, we directly concatenate the knowledge representations with the inputs to each standard transformer head, and then process them using the standard LLM heads. We report the results of the models under different settings in Figure 2. For reference, we also include the results of our method that utilizes the Knowledge Attention mechanism. From the results, we can observe that the model scores under these two alternative settings are lower than those achieved by our proposed method. Possible explanations for this performance difference are as follows: For the first setting, since different knowledge instances are not distinguished, all knowledge instances are treated equally. This lack of differentiation means that potential noise within the knowledge cannot be identified or weighted appropriately, which may mislead the model and result in incorrect answers. For the second setting, the standard transformer heads already focus on processing specific aspects of the input, which do not include the external knowledge we aim to integrate. Therefore, these heads are not adept at handling additional knowledge representations, and cannot efficiently fuse the knowledge into the model's internal representations. This inefficiency leads to a slight decrease in the quality of the generated content. These results further confirm the effectiveness of our proposed knowledge attention mechanism, which allows for a more precise

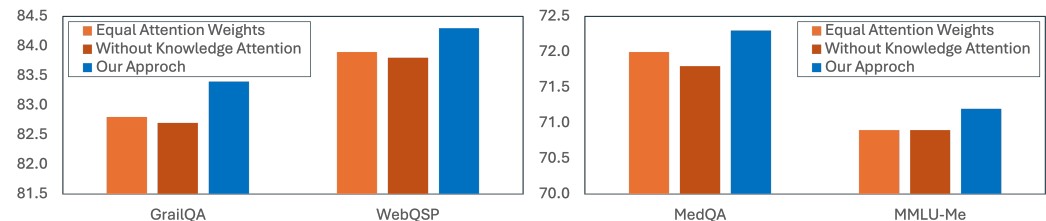

Figure 2: Performance comparison of different knowledge attention settings using LLaMA-2 13B. Our proposed approach with the knowledge attention achieves the best performance by effectively integrating external knowledge.

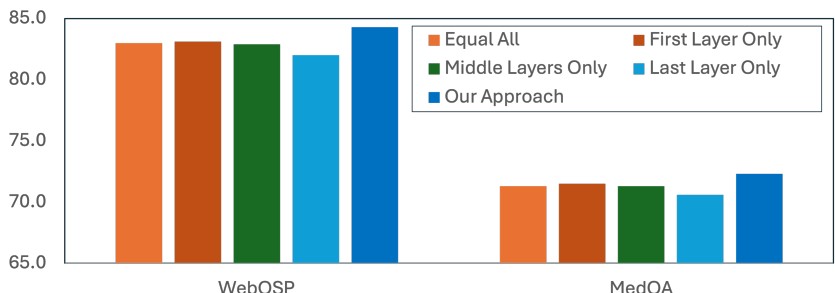

Figure 3: Performance comparison of different knowledge integration layers using LLaMA-2 13B. Our proposed method with the memory module achieves the best performance by dynamically adjusting the integration of knowledge at different layers.

and efficient integration of external knowledge, ultimately improving the model's performance in generating accurate and contextually appropriate responses.

## 4.3 THE EFFECT OF THE LAYER TO INCORPORATE KNOWLEDGE

To investigate the impact of integrating knowledge at different layers of the LLM, we experimented with various settings: (1) **Equal All**: Integrating the knowledge into all layers with equal weights. (2) **First Layer Only**: Incorporating the knowledge at the first layer. (3) **Middle Layers Only**: Inserting the knowledge into the middle layers (e.g., the 20-th layer of the LLaMA-2 13B with 40 layers of multi-head attentions). (4) **Last Layer Only**: Adding the knowledge at the last layer. We present the results of these different configurations using LLaMA-2 13B in Figure 3. For reference, we also include the performance of our method that uses the memory module to dynamically determine the integration weights at different layers.

From the results, we find that the other settings result in inferior model performance compared to our proposed approach. This indicates that it is necessary to differentially integrate knowledge into the LLM at varying degrees across layers. Moreover, we found that incorporating the knowledge into the middle layers yields the best performance compared to integrating at the first or last layers. A possible reason is that the initial layers of the LLM primarily focus on the superficial understanding of the input content, such as lexical and syntactic features (Belinkov et al., 2017), while the last layers are mainly responsible for text generation and high-level abstractions. The beginning and end layers are not primarily engaged in deep semantic understanding or the activation of relevant external knowledge. Therefore, adding knowledge at the initial or final layers may not effectively enhance the model's understanding, leading to suboptimal improvements in performance. In contrast, the middle layers are more focused on deep content comprehension and are thus better suited for integrating external knowledge with the input question. This integration enables the model to utilize the knowledge more effectively, guiding it to generate more accurate and contextually appropriate responses.

## 4.4 CASE STUDY

To demonstrate how our method effectively utilizes knowledge to enhance QA, we present a case study where our approach can produce correct answers. Figure 4 shows the question and the re-

Question

*Which planet in our solar system has the largest number of moons?*
*A. Jupiter*    *B. Saturn*    *C. Uranus*    *D. Neptune*

Attention Weights
| 1 | 2 |
| 3 | 4 |

Knowledge Instances
*1. Jupiter has 79 known moons, including Ganymede, the largest moon in the solar system.*
*2. Saturn has 82 confirmed moons, the most in the solar system as of 2023.*
*3. Uranus has 27 known moons, named after characters from the works of William*
*Shakespeare and Alexander Pope.*
*4. Neptune has 14 known moons, with Triton being the largest.*

Figure 4: An example illustrating how our model leverages the relevant knowledge to produce the correct answer to the question. The average knowledge attentions are visualized, with darker colors indicating higher importance. The correct answer is highlighted in green background color.

trieved knowledge, where the gold standard is highlighted in green. Additionally, we display the average weights assigned to different knowledge instances in our knowledge attention mechanism. In the figure, darker colors indicate higher weights, while lighter colors represent lower weights. From this example, we can see that our approach can effectively identify the most important knowledge for answering the question and utilize it to guide the model to produce the correct answer.

## 5   RELATED WORK

Utilizing knowledge is essential for LLMs to process various tasks, e.g., QA, and thus attracts much attention from existing studies. Some studies directly fine-tune LLMs on data with the required knowledge (Chen et al., 2023; Thangarasa et al., 2023; Sun et al., 2023), yet it is computationally expensive. There are studies that utilize existing resources (e.g., the knowledge base) to improve LLM without updating the LLM's parameters (Baek et al., 2023b; Jiang et al., 2023; Cheng et al., 2023; Wang et al., 2023; Hu et al., 2023). They extract relevant knowledge from a knowledge base and add it to input questions as prompts to instruct the LLM to generate appropriate responses (Asai et al., 2023; Baek et al., 2023b; Fang et al., 2024; Zhu et al., 2024; Xu et al., 2024; Cheng et al., 2024; Jeong et al., 2024; Li et al., 2024). For example, Sun et al. (2023) proposes an approach that enables interactive exploration and reasoning within the graph to enhance inference processes without additional training costs. Jiang et al. (2023) introduces an approach to enhance LLMs' reasoning abilities on structured data by iteratively reading and inferring through a specialized interface, significantly improving performance in low and zero-shot settings. There are other studies that train a small model to better extract and leverage the knowledge (Ma et al., 2023; Wang et al., 2023; Cheng et al., 2024). Compared to existing studies, our approach optimizes the integration of knowledge with LLMs by incorporating the retrieved knowledge into multiple layers of the LLM. We employ a memory module and knowledge attention to differentially fuse various knowledge instances with the LLM, allowing for a more nuanced and effective utilization of external knowledge. This layered and differentiated integration approach enhances the model's ability to generate reliable and accurate answers by selectively emphasizing the most relevant knowledge at each layer.

## 6   CONCLUSION

In this paper, we propose an approach to enhance QA with LLM by retrieval-augmentation using an external attention to incorporate knowledge from certain sources, where such attention serves as an extra head that paralleled and fused with other internal heads from the LLM, so that bringing external knowledge and LLM ability together at the parametric level. Specifically, the knowledge attention encodes and weights knowledge inputs and is incorporated to different layers of the LLM, enabling effective guidance on the generation of accurate and reliable answers. We evaluated our approach on multiple benchmark QA datasets, and the results demonstrate its validity and superiority, where it outperforms strong baselines and existing studies. Analysis on the effect of knowledge attention and how it is fused with the LLM further confirms the effectiveness of our approach.

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
