# OpenReview forum: "Two Heads are Better than One: Retrieval Augmented LLM for Question Answering with External Knowledge Attention"
_ICLR.cc/2025/Conference — ICLR 2025 Conference Withdrawn Submission_

### Official Review · Reviewer_8beM · 2024-10-28

**Soundness:** 2
**Presentation:** 3
**Contribution:** 2
**Rating:** 3
**Confidence:** 4

**Summary:**

The paper proposes a retrieval-augmented question answering model that integrates external knowledge into LLMs through an additional attention head and a memory module controlling knowledge integration across layers. The approach aims to enhance the model's ability to utilize relevant information for answering questions.

**Strengths:**

1. The paper's strength lies in its use of vectorized representations for retrieving knowledge, which allows for a more nuanced integration of external information with the LLM's internal representations. This approach leverages the ability of vector spaces to capture semantic relationships, enabling the model to better fuse and reason over the retrieved knowledge.
2. Another strength is the introduction of a memory module that dynamically adjusts the degree of knowledge integration across different layers of the LLM. This enables more granular control over how external knowledge influences the model's reasoning process, potentially leading to improved performance on complex question-answering tasks.

**Weaknesses:**

1. While the paper presents a novel approach to integrating external knowledge into LLMs, it may lack a broader discussion on the generalizability of this method across different LLM architectures.
2. It is unclear whether the proposed attention mechanism and memory module would require significant retraining or adaptation for each LLM, which could limit its practical applicability.
3. Although the authors explain that this embedding method has advantages (as mentioned in lines 050-085), the article does not explicitly address the robustness of the model to noisy or irrelevant data during retrieval, nor does it provide a comparison with these methods.
* Line 109: There are some errors in the figure references.

**Questions:**

1. Is it necessary to train additional attention and parameter matrices for each LLM to incorporate the proposed knowledge attention mechanism?
2. Regarding noise resistance, how does your method perform compared to other methods under noisy text conditions? For instance, how does the model's performance change when it is fed irrelevant content?
3. Could the authors elaborate on the scalability of the proposed approach, especially when dealing with very large retrieval contexts (beyond BERT's limits) or when the complexity of the questions increases?
4. How does the model handle conflicting information from different knowledge sources, and is there a mechanism in place to resolve or weigh such conflicts?

---

### Official Review · Reviewer_U16r · 2024-11-02

**Soundness:** 2
**Presentation:** 2
**Contribution:** 2
**Rating:** 3
**Confidence:** 3

**Summary:**

This paper focuses on improving the performance of LLMs with RAG on the question-answering (QA) task and proposes an approach to selectively fuse externally retrieved knowledge using different attention mechanisms. Specifically, based on the relevance of the retrieved knowledge, the authors propose encoding and integrating knowledge of varying weights into LLMs at different layers. Experiments on general-domain QA datasets and two domain-specific datasets (medical and counterfactuals) demonstrate improved performance compared to previous baselines. Ablation studies also show the effectiveness of the proposed approach by investigating the knowledge attention weights and the integration of different LLM layers.

**Strengths:**

1. The motivation and idea of improving the integration of externally retrieved knowledge with LLMs in RAG are good and interesting. Indeed, most existing solutions in RAG simply concatenate the retrieved knowledge with the input and send it to LLMs in a shallow way, which may not fully leverage the knowledge and can lead to errors. This paper explores methods to utilize the knowledge in a more manageable way by assigning different attention weights to different pieces of knowledge and integrating them into different layers of the LLM, thereby utilizing the knowledge more deeply.
2. The experiments and ablation studies look sound and can demonstrate the design choices of each component in the proposed method (i.e., knowledge attention, memory module, and fusion in LLM layers).

**Weaknesses:**

1. The usage of notations and symbols in the paper is sometimes misleading, and some of them are not consistent throughout the paper. For example,
- $s_{l,u}$ in equation (7) becomes $s_{l,n}$ in L173
- $f_{K}$ in equation (1) and $f_{KA}$ in L101, L124
- The $H^S$ in equation 3 is different from in L145
- It should be $H^{X}_{l-1,2}$ in equation (9)
2. The datasets and benchmarks used for the general domain QA experiments are somewhat outdated and are not commonly used for evaluating QA in the era of LLMs. This makes the proposed method incomparable to more recent and advanced methods. For example, the community usually uses the following datasets/benchmarks:

 - Natural Questions: A Benchmark for Question Answering Research, Kwiatkowski et al., 2019
 - TriviaQA: A Large Scale Distantly Supervised Challenge Dataset for Reading Comprehension, Joshi et al., 2019
- MuSiQue: Multihop Questions via Single-hop Question Composition, Trivedi et al., 2022
- When Not to Trust Language Models: Investigating Effectiveness of Parametric and Non-Parametric Memories, Mallen et al., 2023
3. The experimental settings are not entirely fair. In Table 4, the backbone models for the baselines are not the same, and some methods are optimized using the training data, while others are not. Furthermore, there is a lot of missing data in Table 4, and the improvement appears marginal compared to Tables 2, 3, and 4, especially considering that the proposed method requires additional training and inference costs. For example, it’s only about a 1-4% improvement compared to the base LLaMA in Table 2.
4. The paper overstates some of its findings and results. For example, it only experiments with the LLaMA model (7B/13B) and BERT as the encoder, yet claims "*our approach works with various pre-trained LLMs*" (L314). In addition, it states in L297 that each model was run three times and that the average and standard deviation were reported, but this information is not shown in the paper.

**Questions:**

1. What is the difference between $f_{KA}$ in L101 and $f_{K}$ in equation (1)?
2. I don’t understand equation 1. Why use $f_K$ to subtract the output of LLM?
3. It appears that the proposed method requires training and parameter tuning on the training and development sets to achieve optimal results for each dataset. How transferable is the proposed method? Will it perform well on other datasets that it has not been trained on?

Typos:
1. L109, Figure ??

---

### Official Review · Reviewer_Gbmc · 2024-11-04

**Soundness:** 2
**Presentation:** 2
**Contribution:** 2
**Rating:** 3
**Confidence:** 4

**Summary:**

The paper introduces a novel method to enhance large language models (LLMs) for question-answering tasks by incorporating external knowledge through a specialized attention mechanism, termed External Knowledge Attention. This approach extends the conventional retrieval-augmented generation (RAG) framework, where relevant external knowledge is retrieved to supplement the LLM’s existing knowledge, thereby improving answer accuracy. The proposed model introduces an additional "external knowledge attention" head that operates alongside the LLM's internal attention heads, enabling a more flexible and dynamic integration of retrieved information based on its relevance to the input query. A memory-based mechanism allows the model to modulate the extent of external knowledge fusion across different LLM layers, adapting based on the relationship between the question and the retrieved content. Experimental results highlight the model's significant performance improvements across both general and specialized question-answering tasks, illustrating its effectiveness in utilizing external knowledge for more accurate responses.

**Strengths:**

The approach of knowledge injection through latent vectors in a decoder-only model is innovative. Most prior studies have focused on incorporating latent knowledge into encoder-decoder models, such as T5, whereas this work applies it to a decoder-only model.

Although the authors do not explicitly address efficiency, the model's use of latent knowledge suggests potential benefits for reducing inference latency and increasing throughput.

**Weaknesses:**

The approach's intuition is not fully explained. The authors argue that standard RAG suffers from "less converged integration of knowledge and LLMs" due to concatenation-based methods, yet this claim lacks citation and evidence. Moreover, the experiments do not directly validate that the proposed approach addresses this issue.

The paper is poorly written, with the methodology lacking sufficient detail. Specific aspects remain unclear, as noted in the questions below.

Certain methodological choices lack justification and intuitiveness, such as the attempt to align the BERT and LLAMA latent spaces without clear post-training evidence.

The approach seems closer to fine-tuning than instruction tuning. Given that LLMs are designed for multiple tasks, the authors should demonstrate that the approach preserves the model's performance on tasks beyond RAG.

**Questions:**

In line 131, the authors mention removing positional embeddings. Could they clarify this choice? Was an ablation study conducted?

How is a single knowledge instance encoded as a single vector? If a pre-trained language model (PLM) like BERT was used, the output would typically be a sequence of latent vectors, not a single vector. The authors should clarify this process.

Why did the authors select BERT specifically? There are more advanced models available, and the choice of BERT requires justification.

[Feedback]
1. Equation 2 has an error; it should be s_u instead of s_i
2. Several recent citations on memory use in RAG settings are missing, such as [1]
[1] https://arxiv.org/abs/2406.04670

---

### Official Review · Reviewer_gndj · 2024-11-04

**Soundness:** 3
**Presentation:** 2
**Contribution:** 3
**Rating:** 6
**Confidence:** 3

**Summary:**

The paper presents an approach to enhance large language models (LLMs) for question answering (QA) tasks through a novel retrieval-augmented generation (RAG) system. This system incorporates external knowledge dynamically using a specialized attention mechanism and memory modules. The proposed method, termed External Knowledge Attention (EKA), integrates an additional "head" within the multi-head attention framework of LLMs, designed to selectively fuse relevant external knowledge into the model’s decision-making process across different layers.

**Strengths:**

1. The introduction of an extra attention head for integrating external knowledge directly within each LLM layer is novel, potentially improving the model's ability to utilize relevant information.
2. The paper provides comprehensive experimental setups and results.

**Weaknesses:**

1. The architecture’s complexity, involving multiple components like memory modules and additional attention mechanisms, might pose challenges in practical implementations or scaling.
2. The multi-hop datasets are simple and small, except for GrailQA.

**Questions:**

1. Can you test your approach on other multi-hop datasets like HotpotQA, Musique, etc?
2. Llama3 was published in early 2024. Can you try to run your experiments on it?
3. Please remove explanations about marks in the caption of Table 5. It is the same as it is in Table 4. The two tables are close enough.
4. Can the knowledge attention module be generalized to fit different scenarios? How is the transferability of the module? Have you tried training this module on one dataset, such as WebQSP, and testing it on another, such as GrailQA?

---

### Note · Authors · 2024-11-18

I have read and agree with the venue's withdrawal policy on behalf of myself and my co-authors.